# A Skin-Conformal, Stretchable, and Breathable Fiducial Marker Patch for Surgical Navigation Systems

**DOI:** 10.3390/mi11020194

**Published:** 2020-02-13

**Authors:** Sangkyu Lee, Duhwan Seong, Jiyong Yoon, Sungjun Lee, Hyoung Won Baac, Deukhee Lee, Donghee Son

**Affiliations:** 1Department of Electrical and Computer Engineering, Sungkyunkwan University, Suwon 16419, Korea; lee39@g.skku.edu (S.L.); dodoworld1993@gmail.com (D.S.); jiyong428@g.skku.edu (J.Y.); hwbaac@skku.edu (H.W.B.); 2Center for Bionics of Biomedical Research Institute, Korea Institute of Science and Technology, Seoul 02792, Korea; leesj9883@kist.re.kr; 3Center for Medical Robotics, Korea Institute of Science and Technology, Seoul 02792, Korea

**Keywords:** fiducial marker, augmented reality, surgical navigation systems, skin-conformal, adhesive patch

## Abstract

Augmented reality (AR) surgical navigation systems have attracted considerable attention as they assist medical professionals in visualizing the location of ailments within the human body that are not readily seen with the naked eye. Taking medical imaging with a parallel C-shaped arm (C-arm) as an example, surgical sites are typically targeted using an optical tracking device and a fiducial marker in real-time. These markers then guide operators who are using a multifunctional endoscope apparatus by signaling the direction or distance needed to reach the affected parts of the body. In this way, fiducial markers are used to accurately protect the vessels and nerves exposed during the surgical process. Although these systems have already shown potential for precision implantation, delamination of the fiducial marker, which is a critical component of the system, from human skin remains a challenge due to a mechanical mismatch between the marker and skin, causing registration problems that lead to poor position alignments and surgical degradation. To overcome this challenge, the mechanical modulus and stiffness of the marker patch should be lowered to approximately 150 kPa, which is comparable to that of the epidermis, while improving functionality. Herein, we present a skin-conformal, stretchable yet breathable fiducial marker for the application in AR-based surgical navigation systems. By adopting pore patterns, we were able to create a fiducial marker with a skin-like low modulus and breathability. When attached to the skin, the fiducial marker was easily identified using optical recognition equipment and showed skin-conformal adhesion when stretched and shrunk repeatedly. As such, we believe the marker would be a good fiducial marker candidate for patients under surgical navigation systems.

## 1. Introduction

With the fast-paced development of medical diagnostic technology in recent times, surgical techniques based on augmented reality (AR) have needed to be introduced for medical procedures. Particularly, surgical navigation systems with optical tracking devices and cameras have the ability to observe affected parts of the human body which can normally not be seen with the naked eye [1,2,3,4,5]. AR-based surgical navigation systems typically use markers and optical tracking devices, such as near infrared (IR) cameras, to visualize affected parts of the body as an image displayed on a monitor to enable more precise surgery (Figure 1a) [6,7]. At present, the fiducial marker uses an optical tracking device to register the position between the patient and the image so as to accurately determine surgical positioning. In this case, metal-based markers are often used for this purpose [8,9,10]. However, these conventional markers are disadvantageous in some cases as they are too rigid, and their temporary adhesion is not good enough during a surgery. These temporary adhesion properties cause markers to slip, particularly if a procedure is postponed with the marker left on the body for extended periods or when the patient suddenly moves during the operation. Should this scenario occur and the fiducial marker misaligns, the exact positioning on the body may be missed entirely, reducing the precision of the surgery, which can ultimately lead to a medical accident. Of the conventional fiducial markers available on the market, no marker possesses a skin-like modulus (about ~150 kPa) nor stretchability (~30%) to permit adhering to any human movements recognizable for diagnostic medical techniques. Several infrared (IR) light-emitting diode (LED)-based fiducial markers have been reported to date. However, almost all are for non-medical purposes or based on rigid substrates. They are neither stretchable nor skin-conformal [11,12,13]. Due to these challenges, we undertook a proof-of-concept study and designed a stretchable electronics system for a new conceptual fiducial marker. Stretchable electronics system means a system consisted of integrated intrinsically stretchable material-based devices, such as LEDs, transistors and sensors, and stretchable interconnect materials on a skin-like rubber substrate [14,15,16,17,18,19,20,21,22,23,24,25,26,27,28,29,30,31,32,33,34,35,36]. However, these devices based on stretchable electronics do suffer from an inherent weakness to external damage and its fabrication process is often difficult and expensive. In order to overcome these problems, researchers have recently been studying polymers with self-healing properties as a potential substrate instead of a rubber [37,38,39,40,41,42,43,44,45,46,47,48,49,50].

In this paper, we introduce a skin-conformal, stretchable, and breathable fiducial marker with human “skin-like” mechanical properties (~150 kPa) and breathability by the adoption of pore patterns. This patch had the ability to maintain its light emission properties when attached to a silicone phantom model and stretched (Figure 1b,c). The patch was also clearly recognized through the near-IR camera and micro-CT machine. It also showed conformal adhesion to the skin when a patient moved. As such, we believe the patch may be a good fiducial marker candidate for patients under surgical navigation systems.

## 2. Materials and Methods

### 2.1. Synthesis of Fiducial Marker

Our fiducial marker was synthesized as outlined in Figure 2a. First, a commercial silicone rubber solution (Ecoflex 00-30, Smooth-on Inc., Easton, PA, USA), employed for its stretchable properties, was mixed and poured into a square petri-dish (12 cm × 11 cm) and then dried and cured for approximately 3 h. After peeling off the substrate, it was then moved into a 10 cm × 10 cm plate, with the edges of the substrate cut to make a square shape. To screen-print the interconnect material, a shadow mask (stainless steel mask, iNEX JK Co., Ltd., Hwaseong, South Korea) was placed on the substrate. After this, a certain amount of conductive silver ink (PE873, Dupont) was collected in front of the exposed pattern area on the mask, screen-printed with a glass slide, and cured (an hour, 120 °C) to create an interconnect for the micro-IR LEDs (QBLP650-IR3, 850 nm, QT Brightek, San Francisco, CA, USA) with soldered wire on their end. The micro-IR LEDs were then attached to the empty site between interconnects on the substrate, and two wires on their ends attached to both ends of the interconnect lines with a silicon adhesive (Silpoxy, Smooth-on Inc.) before being cured at 30 min to fix them. Next, conductive ink was also applied on the interconnect site and cured again. At this stage, the stretchability of the substrate and interconnect were shortly tested manually and quantitively. In manual, they were stretched by two hands. Moreover, a digital multimeter (Keithley 2450 sourcemeter, Tektronix, OR, USA) and motor-based one-axis stretcher (SMC-100, Jaeil optical system, Incheon, South Korea) were used for the cyclic stretching test.

Afterward, the substrate solution was then applied in a way to encapsulate all components of the patch again. Furthermore, the center part of the patch was cut into a square to form a U-shaped patch to be used as a fiducial marker. A suitable adhesive (Silbione RT gel 4717, Elkem Silicones, Oslo, Norway) was then applied to the back of the patch and cured to make an adhesive layer. Finally, pores were drilled using a commercial punch (3 mm-diameter Miltex biopsy punch, Schuco, Watford, United Kingdom) throughout the substrate within 5 mm intervals between the pores, except for parts around interconnects, to give the patch a skin-like low modulus and a breathable property.

### 2.2. Mechanical Characterization of the Substrate and Adhesive

To synthesize the substrate material, we used polydimethylsiloxane (Sylgard 184, Dow Inc., Pittsburg, CA, USA) and Ecoflex (Smooth-on Inc.). Furthermore, the silbione (Elkem Silicones) and polydimethylsiloxane (PDMS) were also used as adhesives. Each was prepared following their technical data specifications. A PDMS was mixed by selected ratios (1:20, 1:30, and 1:40) of a precursor and a curing agent and cured at hot plate. The Ecoflex was mixed with a 1:1 ratio of part A and part B, and then cured at ambient temperature. The silbione was also mixed with a 1:1 ratio of their two precursors and cured on a hot plate. Both the mechanical properties of the substrate, as well as the adhesion energy of the adhesives, were measured using a 90° peeling test from the Instron 900 series tensile test machine [51]. All of the test samples were cut 20 mm × 70 mm in size. Double-sided tape was then attached between the T-shape stage and a slide glass. The front side of the sample (approximately 10 mm) was fixed onto the test jig using PET film and double-sided tape. Samples were then attached onto the slide glass (approximately 30 mm from the back) set at a 25 mm distance between the sample and the end of the test jig. Commercial 3M tape was then attached to the top of the sample to prevent sample delamination while measurements were undertaken. The samples were stretched at a rate of 20 mm per minute to generate a stress–strain curve. A graphic software was used to calculate the adhesion energy of each adhesive. Therefore, the area from 0 to strain value (mm), when force/width (N/m) is the maximum in the stress–strain curve, is the tensile force (Newton, N) applied to a sample that made contact with the glass slide. Finally, the adhesion energy (J/m^2^) was able to be calculated when the tensile force (N) was divided by a thickness of the sample.

### 2.3. Water Vapor Transmission Rate (WVTR) Test

Each sample was tested for its breathability using a well-documented industrial water vapor transmission rate test (ASTM F1249). In this method, dry gas is swept through a chamber where test films act as membranes through the separation of wet and dry gas streams. The partial pressure difference is a driving force for the water vapor to permeate through the film from the high pressure side to the low-pressure side. The barrier of the film then regulates the amount of water vapor transfer; this procedure was performed continuously while being measured by a detector at the outgoing stream of the dry side. 

We, therefore, mimicked this portion of the test to establish the WVTR. In a vacuum chamber, a beaker was filled with hot water for the “wet side” and then covered with both porous and nonporous substrates. The outside of the beaker then acted as the “dry side.” Next, a digital thermo-hygrometer was placed inside the chamber to monitor the humidity in the vacuum for 1 h. The humidity change was measured in real-time every 10 min.

### 2.4. Light Emission and Near-IR Recognition Test

The light emission from the micro-IR LEDs in the fiducial marker was evaluated using a power supply (E3648A Dual Output Power Supply, Agilent Technologies, Inc., Santa Clara, CA, USA). Near-IR recognition was also evaluated by thermal imaging cameras (E75SC, FLIR Systems, Inc., Wilsonville, OR, USA) and near-IR cameras (custom-made, Korea Institute of Science and Technology, Seoul, South Korea).

### 2.5. Demonstration of Skin Adhesion to the Patch

The fiducial marker patch was tested for its skin adhesion properties on a silicone-based phantom model (custom-made, Korea Institute of Science and Technology) and a human back. The patch was firstly attached to the phantom model (connected to the power supply) so that the light of the micro-IR LED light was activated. Patch adhesion was then evaluated by stretching with two hands. Moreover, to evaluate the real skin adhesion, the fiducial marker patch was attached to a human’s back skin. The following 4 movements were repeatedly carried out to evaluate both the adhesion properties of the patch and the light emission characteristics of the micro-IR LED: standing straight, bending the waist forward, pulling two arms to the back, and waist twisting.

## 3. Results and Discussion

### 3.1. Mechanical Characterization and Optimization of Patch Substrate and Adhesive

In order to fabricate a stretchable and adhesive fiducial marker patch, a substrate was selected by comparing the mechanical performance of candidate substrates. First, PDMS 1:20 and Ecoflex were prepared, respectively. The as-prepared sample was then placed onto a glass plate attached to a T-shape stage (Figure 3a). Next, one end of the sample was then fixed to the test jig where the PET film was attached and then measured on a commercial tensile machine at a 90° (Figure 3b,c). The mechanical properties were obtained while the sample was in pull-up mode. Among the substrates, the PDMS 1:20 sample stretched well early on (~50%), but broke shortly after. This was different for the Ecoflex, which had the ability to stretch up to a strain of 300% without breaking (Figure 3d). Next, the PDMS 1:20, PDMS 1:30, PDMS 1:40, and the silbione were then prepared as the adhesives. We also observed their mechanical properties in a similar way (Figure 3e), with the adhesion energies of each material calculated based on the results. Among them, the silbione sample was found with the most superior adhesion energy compared to the other adhesives (Figure 3f). Based on these results, the Ecoflex and silbione were finally selected for the fiducial marker patch. 

We tested the electromechanical coupling of the interconnect material. When the interconnects were stretched to 30%, the LED remained lit (Figure 2b). They also showed stable resistance through cyclic testing (Figure 2c). The resistance was slightly increased, but it was almost stable during the test (approximately under 5 ohm). Therefore, we determined the proper stretchable substrate, interconnect, and adhesive for the best fiducial marker.

### 3.2. Improvement on Skin-Like Properties after the Pore Patterns Adoption

It was important to achieve skin-like mechanical properties. This is to prevent the patch popping out and causing delamination due to the difference in mechanical properties between the skin and the patch if the patient was to move. The patch may also cause patient discomfort or inflammation when attached for long periods of time if the patch is not breathable. To prevent these problems and give the patch a skin-like property, pore patterning was adopted and we compared the properties before and after.

When we compared the strain–stress curves of nonporous (NP) and porous (P) substrates, the porous substrate showed lower modulus under 100 kPa at 150% strain in comparison to the nonporous substrate, which is closer to the physical properties of human skin (Figure 4a). The adhesion energy, however, was found to only slightly decrease for the porous substrate due to the reduction in surface area by pore patterns (Figure 4b). The water vapor transmission rate (WVTR) was then tested using hot water and implemented simply to investigate the breathability of the two substrates. Figure 4c shows the results of the relative humidity on the dry side of the chamber, which was found to sharply increase on the porous substrate for 30 min. 

In contrast, the nonporous substrate was not able to transmit steam from the “wet side” of the beaker to the “dry side.” Figure 4d,e shows that the porous substrate did in fact transmit the steam at a much faster rate. Through this, we concluded that the pore patterns not only give the patch a skin-like modulus but also breathability expressed by vapor transmission, so it makes the patch more comfortable for the patient’s skin.

### 3.3. Light Emission and Near-IR Recognition Test

Generally, fiducial markers should be responsive to the two light sources of the optical tracking device and the C-shaped arm (C-arm) in the surgical navigation systems to display the affected area. Typical optical tracking devices often use 850 nm near-IR cameras that recognize reflection patterns through a difference in the near-IR absorbance, and thus noninvasively displaying the position of the affected part in real-time. The C-arm, which is commonly used as a CT imaging device, visualizes the affected part via X-rays. In order to use the patch as a fiducial marker, it must be recognized on these devices. Specifically, the fiducial marker generally uses an optical tracking device, which serves to define the patient’s coordinates with the marker in surgical navigation systems, so it must operate at near-IR light. For the use as a fiducial marker, we tested the light emission and wavelength compatibility (i.e., 850 nm) of the six micro-IR LEDs attached to a patch. We first tested whether or not the LEDs connected to the stretchable interconnects emitted light. The LEDs for all the patches (n > 10) were found to emit light within a threshold voltage of 7–8 V. We also verified the light response of the patches, which maintained a voltage of 11 V (Figure 5a). To evaluate the compatibility and selective resolution for near-IR rays, images were obtained at each voltage using a thermal imaging camera and a near-IR camera at all wavelengths (Figure 5b,c). The patch was found to show no light emission below 7 V. On the contrary, at 9 V and above, the brightness was too high, so the individual spots could not be distinguished. From these results, we confirmed that the optimal near-IR resolution voltage of the patch was around 8 V, which was typically where the LED size and the light source size were in a 1:1 ratio. Furthermore, the marker was also shown to be selectively observed when it was analyzed with the near-IR camera as compared to the thermal imaging camera. Micro-CT imaging was also used to verify the X-ray stability of the LEDs (Appendix A). We confirm that the LEDs used in the patch could operate without any damage when exposed to the X-ray. This then proves that the patch has good responsiveness and stability at near-IR and X-ray wavelengths and can be sufficiently used as a fiducial marker for patients under surgical navigation systems. 

### 3.4. Skin-Conformal Adhesion of the Patch

Currently, surgical navigation systems are mainly used for spinal surgery. The patch was also developed with spinal surgery in mind, so the direct attachment to the back of a human subject was evaluated to test its adhesion to real skin. During this evaluation, the person used for the study was asked to stand straight (Figure 6a), bend his waist forward (Figure 6b), pull two arms back (Figure 6c), and to twist his waist (Figure 6d) repeatedly. As a result, the subject felt no discomfort caused by the rigidity of the LEDs when the patch was attached to the back. It was proved that the power was stably supplied by the stretchable interconnects in the patch because the LEDs on the patch were not turned off while four different movements were implemented. This result indicates that the patch could be a useful fiducial marker for patients due to its advantageous skin-conformal adhesion properties, even when the patient moves.

## 4. Conclusions 

In this study, we optimized the proper materials to create a fiducial marker patch with a low modulus and excellent adhesion energy. We adopted pore patterning to ensure the patch had a skin-like mechanical property and breathability. The IR-LEDs built into the patch clearly displayed the attached area within the thermal imaging camera, near-IR camera, and micro-CT. When the patch was attached to a human back, we found that the skin-conformal adhesion and continuous light emission stability of the patch persisted while various movements were undertaken by the subject. The as-synthesized patch was also not rigid because of its rubbery substrate and soft adhesive, so it showed good skin-conformal adhesion to a human’s back skin. Therefore, we expect that the patch will be a new candidate as a skin-conforming, stretchable, and breathable fiducial marker patch for surgical navigation systems.

## Figures and Tables

**Figure 1 micromachines-11-00194-f001:**
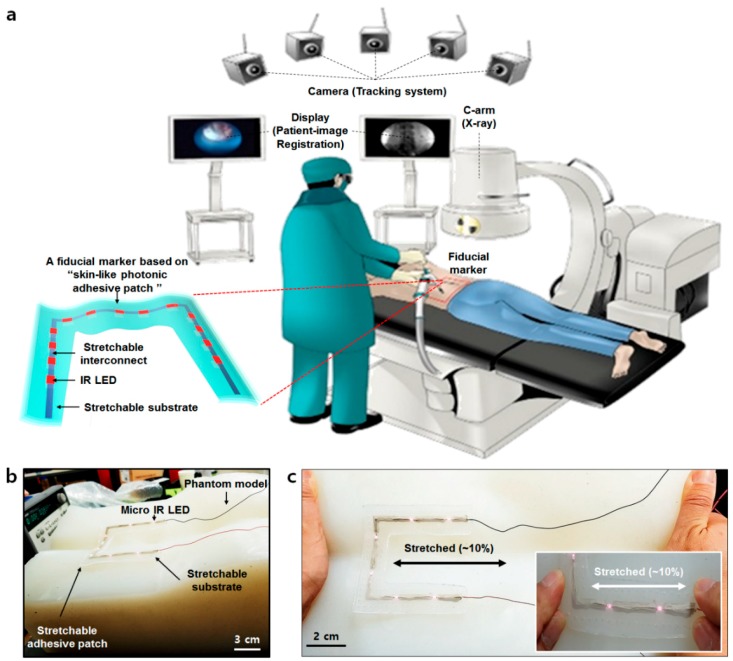
Schematic of a surgery navigation system and skin-conformal fiducial marker patch. (**a**) Schematic of an overall surgery navigation system and its components. (**b**) The components of the fiducial marker patch. (**c**) The stretchability of the fiducial marker patch.

**Figure 2 micromachines-11-00194-f002:**
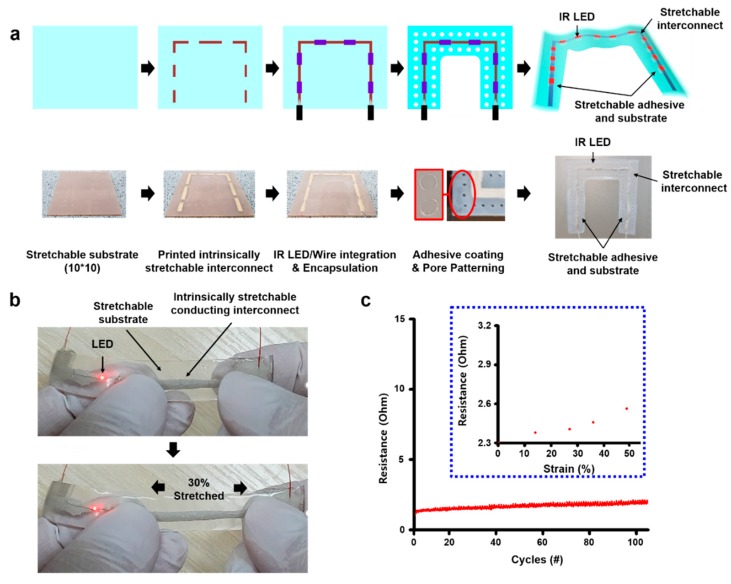
Marker patch fabrication process and the stretchability of interconnect material on the patch. (**a**) Patch fabrication process. (**b**) The stretchability of the patch interconnect part with micro-IR LEDs when it was stretched to 30% by two hands. (**c**) Electrical stability of the interconnect part by 100 cycle stretching test. Inset: the resistance change of the interconnect part by strain.

**Figure 3 micromachines-11-00194-f003:**
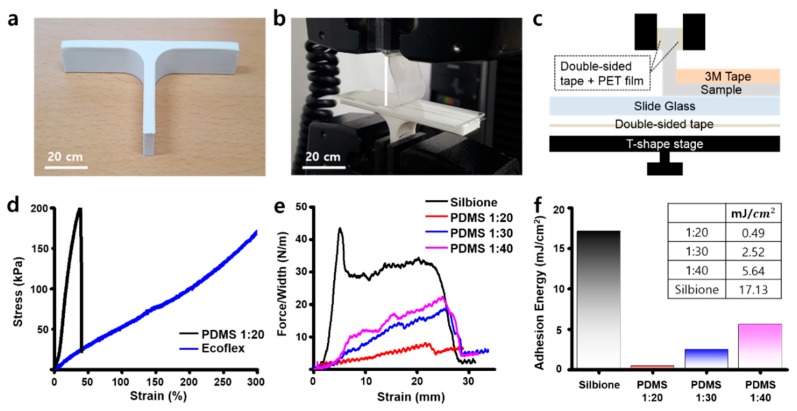
Mechanical characterizations of the patch components. (**a**) 3D-printed T-shape stage for the adhesion test. (**b**) The adhesion test using T-shape stage and tensile machine. (**c**) The components of the adhesion test system. (**d**) The stress–strain curve of substrate materials. (**e**) The stress–strain curve of adhesives. (**f**) The adhesion energy calculation results of adhesives. Inset: the adhesion energy of them.

**Figure 4 micromachines-11-00194-f004:**
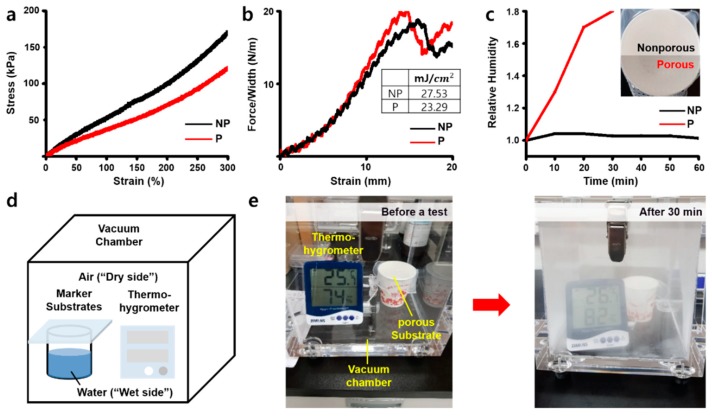
Mechanical characterizations of the skin-like porous patch. (**a**) The stress–strain curve of both nonporous (black) and porous substrate (red). (**b**) The stress–strain curve of adhesives on both nonporous (black) and porous substrate (red). Inset: the adhesion energy of them. (**c**) The relative humidity of both nonporous (black) and porous substrate (red). Inset: the image of nonporous (top) and porous substrate (bottom). (**d**) The structure of water vapor transmission rate test chamber inspired by ASTM F1249. (**e**) Before and after the water vapor transmission rate test of the porous substrate.

**Figure 5 micromachines-11-00194-f005:**
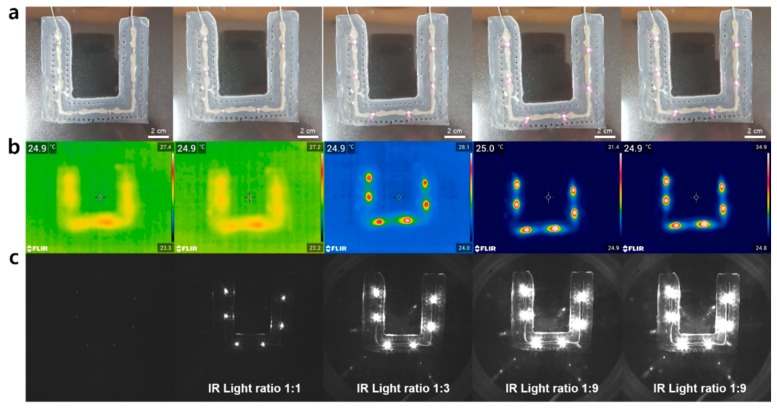
Light emission and near IR recognition test of the patch. (**a**) Images of the fiducial marker patch. (**b**) Thermal imaging camera images of the patch. (**c**) Near-IR camera images of the patch.

**Figure 6 micromachines-11-00194-f006:**
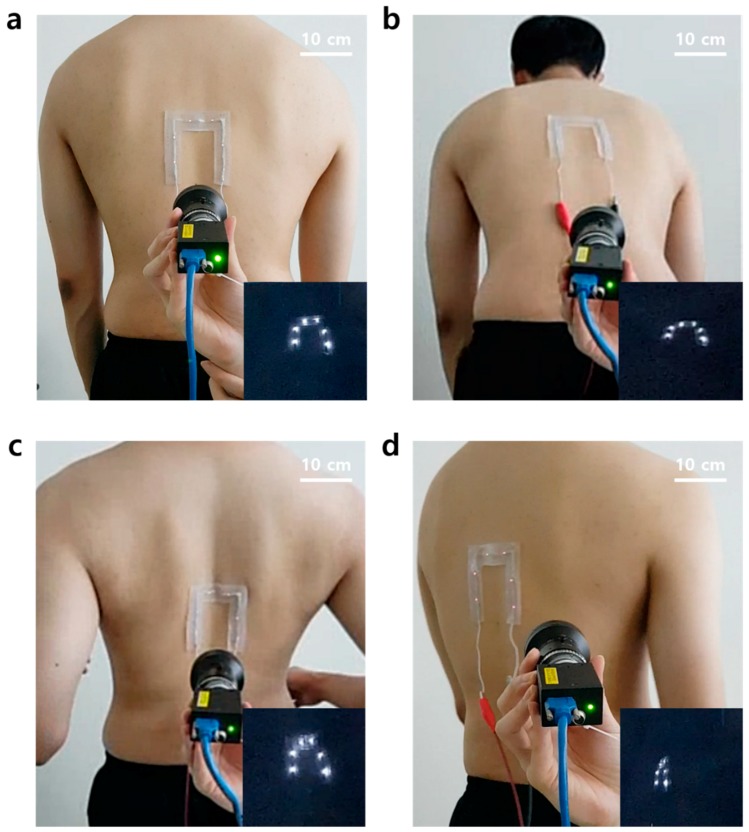
Skin-conformal adhesion demonstrations of the patch. (**a**) Standing straight. (**b**) Bending the waist forward. (**c**) Pulling two arms to the back. (**d**) Twisting the waist.

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
