# Peer review of "A Skin-Conformal, Stretchable, and Breathable Fiducial Marker Patch for Surgical Navigation Systems"

_micromachines, 2020, doi:10.3390/mi11020194_

Round 1
Reviewer 1 Report
The authors reported stretchable, breathable fiducial markers with micro-IR LED's, which is an interesting study to the stretchable community.
Below are my minor comments.
1. “a certain amount of conductive 88 ink (Ag ink, Dupont)”
-> Could the authors kindly write down the product number of the Ag ink so that it could be informative to other researchers?
2. “Finally, pores are drilled throughout the substrate, except for parts around interconnects, to give skin-like low modulus and breathable property of the patch (3mm)”
-> What tool did the authors use to drill pores? What is the size of a pore? What is the spacing between pores?
3. Could you also provide the product and company name of micro IR LED’s?
Reviewer 2 Report
In this work, the authors use Ag ink connected to microLEDs atop silicone substrates to fabricate a fiducial marker that could be use in surgery. In my opinion, while the research displays a great example of one potential application for stretchable electronics, the work is not a substantial achievement for stretchable electronics. Since I do not have expertise in healthcare, I do not find it appropriate to determine whether the progress in this area warrants publication. For example, in reading about fiducial markers, I have learned that fiducial marker placement is very important for radiation therapy. Would these devices be used for radiation therapy? If so, how would radiation interact with the traces and markers? Or would they only be used for triangulation?
Then, I would wonder whether the work is appropriate for micromachines since the device is fairly large and is not really a machine. At the very least, I would say that the work could do with major revisions. In addition to changes outlined below, I have a few specific suggestions/questions.
There were many typos/confusing phrasings. I suggested some changes to help the authors, but I am sure that I missed some.
The authors introduce pores to the substrate to improve breathability. I wonder what the minimum surface area is needed for adhesion. The photograph of the patch has an excess of insulating substrate that could be cut away, leaving only the wires atop the skin. This would be substantially more breathable than the patch with pores. If the wires are adhesive, the pores would be unnecessary.
The authors argue that current fiducial markers can move when the patient moves. It appears that the markers may still move when the patient moves in Figure 6. The question would be whether they are in the same place after the patient is done moving. The authors should have a “patient” perform a single movement multiple times while the IR camera is on a tripod. Then the positions of the markers before and after movement (perhaps relative to one another) could be monitored to see if the patch slips during movement.
I would recommend changing the title to “skin-conforming” not “skin-conformal”. This may not be a necessary correction, just a recommendation. “Skin-conforming” would mean that it conforms to the skin. “Skin-conformal” just seems weird to say, although I don’t know if it is wrong since there are examples of others using it in the literature. Should change throughout the manuscript if changed.
I hope the authors find the suggestions helpful while they improve their manuscript.
Line 18: “evaluate” to “in evaluating” Question: do these navigation systems actually evaluate the location and impacts of ailments? Or just help with surgery? Why would you start surgery without knowing location/impacts of ailments? This might be one aspect of the paper that I do not have the expertise to evaluate properly.
Line 21: “operators using multifunctional” to “operators using a multifunctional” – this sentence is generally confusing – should reword “These markers then guide operators who are using a multifunctional endoscope apparatus by signaling the direction or distance needed to reach affected parts of the body.”
Line 22: confusing use of “they” – the operators? The apparatus? – the correct word is most likely “fiducial markers” but even then I’m not sure.
Line 26: “mechanistic” when meant “mechanical”
Line 30: “yet” to “and” – yet indicates breathable is normally excluded but there is no reason to believe that.
Line 33: for clarity: “equipment, showing” to “equipment and showed”
Line 34: “for patients under surgical” to “for patients utilizing surgical”
Line 41: “medical procedures, particularly…” to “medical procedures. Particularly , surgical navigation systems with optical tracking devices and cameras have the ability to observe….”
Line 49 “10], however, are” to “10]. However, these markers are”
Line 50: I think you’d want the markers to be temporary adhered – the other option would be permanent. I would guess this can be reworded to say that the temporary adhesion is not good enough, which is what the next sentence says. But this should be reworded to not make it seem like permanent would be better.
Line 53: “surgery that” to “surgery, which”
Line 55: “kPa) or a stretchability (~30%), can adhere” to “kPa) nor stretchability (~30%), to permit adhering”
Line 59: “an” to “a”
Line 62: I don’t think the LEDs that you are using are stretchable themselves but just have stretchable interconnects.
Line 63: “so have” to “so they have”
Line 64: “skin, having excellent” “skin and have”
Line 64: Do these systems of biosensing capabilities? It doesn’t seem true based on the architecture. The authors should include a relevant citation if they are referring to the general literature.
Line 66: The authors should provide a citation for “difficult and expensive” procedures for related stretchable electronics. There are many implementations similar to what the authors are proposing that are not costly nor difficult to fabricate.
Line 71: “silicon phantom model” to “silicone phantom model”
Line 88: “was was” to “was”
Line 95: “And a” to “A”
Line 104: (Figure 2) The authors show a continuous curve from 0 to 50 % strain with sharp points at about 15% and about 30%. It makes me wonder if there are just ~5 or so data points, or what might be causing the discontinuity.
Related to the above figure, it is surprising that the resistance does not follow Ohm’s law when stretched (decreased cross section area and increased length should increase resistance much more substantially – to the square of the elongation). It’s noted that in Figure 2b there are connecting wires. Does this electromechanical measurement use a 2-point probe or 4-point probe measurement? Are the wires that connect to the stretchable conductor as shown in 2b being included in the resistance measurement? It’s possible those wires limit the current.
Line 111: “And the silbione” to “Silbione”
Line 113 “A PDMS is” to “PDMS was”, “cured at” to “cured on a” – What temperature was the hot plate?
Line 122: It’s unclear what the purpose of the 3M tape is. Don’t you want delamination so you can measure adhesion properties of the ecoflex rather than the stress-strain characteristics? Are the stress strain characteristics in 3d from this measurement? This seems like a roundabout and possibly unreliable way of measurement when you can just use clamped samples.
Line 123: “The tensile test started…” to “The samples were stretched at a rate of 20 mm per minute to generate a stress-strain curve.”
Line 125: “And graphic software” to “Graphic software”
Line 126: “The” to “the” – This sentence is also confusing. Unclear what is meant by area under the curve being equated with tensile force.
Line 133: “A” to “a”
Line 151: “4 movements was” to “4 movements were”
Line 169: It’s unclear why PDMS was not good enough as a substrate (since the body may not need deformations of >50% strain). The use of PDMS as adhesion layers is also unclear. How would the adhesion of PDMS be significantly different from that of Ecoflex? It would make sense that silibione would be the best adhesion layer.
Line 170: Figure 2e – the x-axis indicates strain but would be more accurately described as extension unless the units of mm are incorrect. Figure 2f – the adhesion energy is in J/cm2 on the axis but in mJ/cm2 in the table in the inset of the figure, with the values being identical.
Line 176: “And, we shortly an electrical stability” to “We tested the electromechanical coupling”
Line 177: “When they were stretched to 30% by two hands, the LED light was kept” to “When the interconnects were stretched to 30%, the LED remained lit.” – “And they” to “They” – “showed their stable electrical performance by the cyclic stretching test” to “showed stable resistance through cyclic testing”
Line 178: “Their resistance” to “The resistance”
Line 179: I wouldn’t say that this implementation would be the “best” set of materials since the authors only optimized within a constrained subset of materials.
Line 182: Remove “To increase skin-like properties of the patch”
Line 184: “They may…” to “The patch may also cause patient discomfort or inflammation when attached for long periods of time if the patch is not breathable.”
Line 187: “and compared it properties” to “and we compared the properties”
Line 190: There are many values given for human skin – e.g. ,a quick google search indicates between 420 kPa and 850 kPa. To state that the porous structure is closer to human skin seems erroneous. In addition, both samples show moduli under 100 kPa.
Line 198: Figure 4d does not appear to show anything about the porous substrate transmission.
Line 216: How does the presence of the conductive Ag traces influence the ability to properly image with CT imaging? Are they thin enough to be invisible to the CT or do they block a significant enough region to cause problems?
Line 222: remove “actually”
Line 226: remove “And”; Unclear what is meant by “light size” – this sentence should be revised for clarity. E.g., “At 9 V and above, the brightness was too high and the individual spots could not be distinguished” or something similar that is still accurate
Line 230: “To be selectively…” – something appears to be missing here.
Line 242: “bended” to “bend”
Line 244: remove “And”
Line 253: Again, I would not say that this is the best, but it might be an improvement relative to other techniques.
Line 254: remove “And”, “we adopted a” to “we adopted”
Line 255: “and the” to “and”
Line 258: “of the patch while” to “of the patch persisted while”
Line 259: “substate” to “substrate”
Line 260: confusingly worded sentence
Line 261: “Therefore, we expected the patch…” to “Therefore, we expect that the patch will be a new candidate as a skin-conforming, stretchable, and breathable fiducial marker patch for surgical navigation systems.”
Reviewer 3 Report
In this manuscript, the authors developed the stretchable patch integrated with interconnects and LEDs for the application to fiducial markers on the skin. Utility of the patch is clear. The physical properties of the patch are well characterized.
(1) Breathability of the developed patch appears to be insufficient, because pores are macroscopic and they do not uniformly cover the substrate. The pores need to be more densely placed on the entire surface of the stretchable substrate. I am not confident that ASTM F1249 is an appropriate method for this kind of substrate with macroscopic pores. An example paper that could be compared with the present manuscript is Miyamoto, Lee, et al., Nature Nanotech 12, 907–913 (2017) (DOI:10.1038/nnano.2017.125), where the pores are microscopic and uniformly exist on the substrate.
These points have to be discussed in the manuscript, as well as the size, positions, and density of pores need to be described.
(2) Grammatical error: line 190: more lower modulus -> lower modulus. I recommend the authors check the entire text again.
Round 2
Reviewer 2 Report
After re-reading the manuscript, I am not quite convinced by the implementation. I really hope that something like this would be useful for surgical devices and related technologies, but I do not feel that there is any new scientific insight brought about by the work. The methodology for synthesis of the composite is very straightforward, and the characterization is consistent with many other reports of similar materials.
The implementation into a surgical device might be interesting, but I don't have the appropriate background to evaluate how novel it is. The authors claim that metal interconnects are typically used and they are too rigid, but I could see the same architecture being used with bent wires, which would be very inexpensive and not require composite processing. I would defer to other reviewers on this aspect of the work. Based on the stretchable electronics aspect of the work, I can't recommend for publication but am not advising that it shouldn't be published - just that the stretchable electronics contribution is not impactful.
I do have a few replies to the authors' rebuttals:
Comment #3: The authors introduce pores to the substrate to improve breathability. I wonder what the minimum surface area is needed for adhesion. The photograph of the patch has an excess of insulating substrate that could be cut away, leaving only the wires atop the skin. This would be substantially more breathable than the patch with pores. If the wires are adhesive, the pores would be unnecessary.
Our response: The wire part (stretchable conductor) is stiff after it is cured. Their modulus is bigger than the skin (~150 kPa). As the reviewer mentioned, if we cut out the excess pore area from the patch and leave only the wire part, stiffness of the interconnect is increased, finally causing undesired deformation and delamination. In other words, the high modulus of the interconnect will be more dominant when the excess area of the substrate is removed. Thus, we chose to give the patch enough area and introduce the pores for reduction of overall stiffness. Figure 4a supports that the application of pore to the patch is highly effective in making the patch skin-conformal.
New reply: The main argument for why these patches are better than current markers is that they are softer. But if the conductive wires are too stiff without the excess insulating rubber, how is that situation different than using a thin conducting copper wire in a serpentine pattern with a simple fabrication technique (e.g., formed by bending the wire by hand and curing the wire in rubber)? The stretchable electronics aspect of the work does not strike me as compelling.
Comment #4: The authors argue that current fiducial markers can move when the patient moves. It appears that the markers may still move when the patient moves in Figure 6. The question would be whether they are in the same place after the patient is done moving. The authors should have a “patient” perform a single movement multiple times while the IR camera is on a tripod. Then the positions of the markers before and after movement (perhaps relative to one another) could be monitored to see if the patch slips during movement.
Our response: Silbione was used as an adhesion material in stretchable/wearable devices (Jang, K.I. et al, Rugged and breathable forms of stretchable electronics with adherent composite substrates for transcutaneous monitoring, Nature Communications, 2014, 5, 4779). From the reference, silbione-coated soft and stretchable substrates can be mounted well without delamination from skin even when strained. As the reviewer pointed out, in the future, we will make the stretchable substrate more thin than before to be conformally mounted on skin.
New reply: This experiment that was suggested does not seem difficult to do. Since the stretchable electronics aspect of the work is not too impactful (and could possibly be replaced with other materials), it would be worthwhile to try to demonstrate that the markers are state-of-the-art.
Comment #6: The authors show a continuous curve from 0 to 50 % strain with sharp points at about 15% and about 30%. It makes me wonder if there are just ~5 or so data points, or what might be causing the discontinuity. (Line 104, Figure 2)
Related to the above figure, it is surprising that the resistance does not follow Ohm’s law when stretched (decreased cross section area and increased length should increase resistance much more substantially – to the square of the elongation). It’s noted that in Figure 2b there are connecting wires. Does this electromechanical measurement use a 2-point probe or 4-point probe measurement? Are the wires that connect to the stretchable conductor as shown in 2b being included in the resistance measurement? It’s possible those wires limit the current. (Figure 2c)
Our response:
Firstly, we changed Figure 2c to avoid the confusion between the cyclic test graph and inset. The inset in Figure 2c shows almost linearity. The small changes of the slope in a specific range originate from the mechanical modulus mismatch between the interconnect and substrate. The mismatch results in deformation (stretched and twisted on the stretched substrate) of the interconnect when stretched. The physical deformation of the stretched interconnect causes small changes of its resistance values
For measuring conductivity of the stretchable electrode, 4-point probe measurement is required. However, in this study, we measured the resistance of the interconnect using 2-point probe because it is more important that delivering stable power to IR LEDs without analyzing or controlling electrical percolative pathways in the stretchable polymer matrix. Considering our IR LED operation, electrical property of the interconnect is acceptable.
Our modification to the figure:
(Figure 2c in revised Main Text)
The inset graph on Figure 2c is changed but there is no change on the caption of this figure.
Figure 2. (c) Electrical stability of the interconnect part by 100 cycle stretching test. Inset: The resistance change of the interconnect part by strain.
New reply: The inset still does not follow Ohm’s law. At 50 % strain, you would expect at 2.25 x increase in resistance. It would be important to mention in the caption or manuscript that the measurement includes resistance from connecting wires, which is why the resistance change is so small relative to the total resistance.